Female philopatry may influence antipredatory behavior in a solitary mammal

Burnett Alexandra 1 aburnett93@arizona.edu
Hein Michelle 2
Payne Natalie 1
http://orcid.org/0000-0003-2635-4782 Vargas Karla L. 1 3
http://orcid.org/0000-0001-5380-3059 Culver Melanie 1 4
http://orcid.org/0000-0003-1406-9853 Koprowski John L. 1 5
1 School of Natural Resources and the Environment, University of Arizona , Tucson, AZ , United States
2 Ecology and Evolutionary Biology, University of Arizona , Tucson, AZ , United States
3 The Biodesign Center for Fundamental and Applied Microbiomics, Center for Evolution and Medicine and School of Life Sciences, Arizona State University , Tempe, AZ , United States
4 U.S. Geological Survey, Arizona Cooperative Fish and Wildlife Research Unit, University of Arizona , Tucson, AZ , United States
5 Haub School of the Environment and Natural Resources, University of Wyoming , Laramie, WY , United States
Sunny Armando
Electronic publication date: 2025 Mar 20
Publication date: 2025
Volume: 13
Electronic Location ID: e18933
Received 2024 Sep 6; Accepted 2025 Jan 14
Copyright: © 2025 Burnett et al.
Copyright year: 2025
Copyright holder: Burnett et al.
License: This is an open access article distributed under the terms of the Creative Commons Attribution License, which permits unrestricted use, distribution, reproduction and adaptation in any medium and for any purpose provided that it is properly attributed. For attribution, the original author(s), title, publication source (PeerJ) and either DOI or URL of the article must be cited.
License URL: https://creativecommons.org/licenses/by/4.0/

Keywords: Ammospermophilus harrisii, Philopatry, Kin selection, ddRADseq, Antipredator, Behavior

Funding: T&E, Inc Support for this project was received from T&E, Inc. The funders had no role in study design, data collection and analysis, decision to publish, or preparation of the manuscript.

==============================
Whether neighboring individuals are related or not has a number of important ecological & evolutionary ramifications. Kin selection resulting from philopatry can play an important role in social and antipredatory behavior. Ground squirrels exhibit alarm vocalizations in the presence of predators; however, the degree to which kin selection shapes alarm calling behavior varies with species ecology and the degree of relatedness between neighbors. We studied a solitary ground squirrel species that exhibits sex-biased calling propensity to determine if female philopatry may be responsible for sex differences in antipredatory behavior observed in our population. We used double digest restriction-site associated DNA sequencing (ddRADseq) to sample the genomes of Harris’s antelope squirrel (Ammospermophilus harrisii) to determine the relatedness between individuals and test whether genetic and geographic distance were correlated. We found that geographic distance had a positive relationship with genetic distance, and that this relationship was sex-dependent, suggesting male-biased dispersal. Our results provide supporting evidence that female philopatry may be responsible for higher calling propensity observed in female squirrels, potentially influencing antipredatory behavior in this species via kin selection. Our findings add to a growing body of evidence that philopatry is an important ecological driver influencing sociospatial organization in solitary species.

Introduction

Relatedness between neighboring conspecifics has a number of ecological and evolutionary consequences (Lawson Handley & Perrin, 2007; Clutton-Brock & Lukas, 2012; Nichols, Zecherle & Arbuckle, 2016), affecting fitness (van Noordwijk et al., 2012; Jungwirth et al., 2023; Walmsley et al., 2023) and behavior (Thorington et al., 2010; Williams et al., 2013), including spatial and social organization (Støen et al., 2005; de Oliveira et al., 2022; Ausband, 2024). High relatedness between neighbors often results from philopatry, in which adults settle close to their natal home range, and dispersal is often sex-biased (Lawson Handley & Perrin, 2007). Mammals typically exhibit male-biased dispersal (Lawson Handley & Perrin, 2007; Mabry et al., 2013), and daughters can benefit from remaining close to their mothers through behaviors such as inheriting territory (Støen et al., 2005; Goodrich et al., 2010) or breeding position (Ausband, 2024). Antipredatory behavior can also be affected by sociospatial patterns resulting from philopatry and may represent an additional benefit of philopatry. Alarm vocalizations may benefit neighboring conspecifics by functioning as a warning that a predator is present, directly deterring predators, or both (Sherman, 1977; Schel, Candiotti & Zuberbühler, 2010; Fuong, Maldonado-Chaparro & Blumstein, 2015; Isbell & Bidner, 2016). Alarm calling behavior can therefore be influenced by kin selection where species live within close proximity of relatives (Dunford, 1977; Sherman, 1977; Wheeler, 2008).

Ground squirrels of the family Sciuridae emit alarm vocalizations in response to predators, and many ground squirrels live in close proximity to their relatives. Kin selection is therefore thought to be a strong evolutionary driver of calling behavior (Dunford, 1977; Sherman, 1977). Male ground squirrels often disperse away from natal burrows whereas females typically exhibit philopatry (e.g., Holekamp, 1984; Shriner & Stacey, 1991; Neuhaus, 2006; Nguyen & Van Vuren, 2024). As a result, adult males may be unrelated to surrounding conspecifics, so alarm calling would not benefit adult males via indirect fitness. Thus, individuals emitting alarm vocalizations may be more likely to be female in species with male-biased dispersal (Dunford, 1977; Sherman, 1977). The rate of male-biased dispersal can vary substantially across ground squirrel species, however, with more social species exhibiting higher rates of male dispersal (Devillard et al., 2004). Further, not all squirrel species exhibit close relatedness between neighboring squirrels (Munroe & Koprowski, 2014; Glover, 2018). For example, the round-tailed ground squirrel (Xerospermophilus tereticaudus) lives gregariously in groups of unrelated individuals (Munroe & Koprowski, 2014). Round-tailed ground squirrels also emit alarm signals, but because they live in groups of unrelated individuals, kin selection is not likely to be a primary driver of alarm calling behavior. Other evolutionary drivers of alarm vocalizations could include direct fitness (Sherman, 1977; Blumstein et al., 1997) or reciprocity (Trivers, 1971, but see discussion in Blumstein, 2007). Alarm calls may also be directed at predators (Wheeler, 2008; Digweed & Rendall, 2009; Burnett & Koprowski, 2020), and predator-prey interactions could be the primary driver of alarm calling behavior in some squirrel species.

The Harris’s antelope squirrel (Ammospermophilus harrisii) is a solitary species that emits alarm calls throughout the year, regardless of juvenile presence, indicating that alarm calls may be directed toward predators as a deterrent (Burnett & Koprowski, 2020). Opportunistic observations of alarm vocalizations showed a higher proportion of female callers (Burnett & Koprowski, 2020), suggesting that calling behavior could be influenced by female philopatry and kin selection. Although adult A. harrisii live solitarily in large home ranges, neighboring home ranges overlap (Burnett & Koprowski, 2024). Further, A. harrisii alarm vocalizations can be high amplitude (A. Burnett, 2017, personal observation) and are structured to carry over long distances (Bolles, 1988), potentially benefiting neighbors if individuals use alarm vocalizations as a warning or if predators are deterred from the area. However, whether neighboring A. harrisii are related is unknown. To understand whether philopatry could influence calling propensity in A. harrisii, we sampled the genomes of neighboring individuals to analyze their relatedness and estimate any sex-dependency. Given our observation of more female antelope squirrels alarm calling (Burnett & Koprowski, 2020), we expect patterns in relatedness of A. harrisii to be indicative of female philopatry, in which female-female relatedness is higher than male-male relatedness and the relationship between genetic and geographic distance is sex-dependent, showing relatedness between neighboring females but not males. Conversely, if neighboring individuals are not related or dispersal is not male-biased, higher call propensity in females may be maintained primarily through direct fitness or alternative selection pressures (e.g., Blumstein et al., 1997).

Materials and Methods

Sample collection and DNA extraction

We sampled A. harrisii in the Santa Rita Experimental Range as part of a larger research study conducted in 2017 and 2018, using the same focal areas as those in Burnett & Koprowski (2020) in which squirrels were observed to exhibit sex-biased calling. We chose to sample in the Santa Rita Experimental Range because we were specifically interested in whether the sex-biased calling observed in this population could be influenced by relatedness between individuals, and whether other populations also exhibit sex-biased calling has not yet been investigated. We focused our trapping efforts around four focal areas, where we experienced much higher trapping success compared to other locations in the study area (Burnett & Koprowski, 2020, 2024). Harris’s antelope squirrels were baited and captured with Tomahawk live traps (model No. 201; Tomahawk Live Trap, Hazelhurst, WI, USA) in the Santa Rita Experimental Range (SRER), located in the Sonoran Desert approximately 65 km south of Tucson, Arizona (Fig. 1). Traps were checked frequently (once/hour) and shaded with vegetation and/or shade cloth to prevent heat stress and sun exposure. Once captured, we used a cloth handling cone (Koprowski, 2002) to minimize stress during handling. We recorded weight, sex, life stage, and reproductive status for each animal captured and collected ear tissue samples with an ear punch (Fisherbrand Animal Ear Punch, 1 mm, Thermo Fisher Scientific, Waltham, MA, USA). Squirrels were tagged with sterile passive integrated transponder (PIT) tags (HPT9, 8.4 × 1.4 mm, 0.02–0.04% body weight, Biomark, Inc., Boise, ID, USA) before being released. Some adult individuals (>110 g) were additionally fitted with a radio collar for VHF tracking (Wildlife Materials; <5% body weight; see Burnett & Koprowski, 2024). We took no more than one tissue sample per ear from squirrels captured more than once (n = 47 individuals, 51 tissue samples prior to removal of duplicate individuals). We received approval from University of Arizona Institutional Animal Care and Use Committee (16–169) and complied with the Animal Welfare Act for all procedures. We additionally followed ethical guidelines for trapping and handling small mammals published by the American Society of Mammalogists (Sikes & Animal Care and Use Committee, 2016). We obtained a scientific collecting permit from Arizona Game and Fish Department (SP501610 & SP611944). We did not give the animals anesthesia or analgesia because the effects of these agents are not well-studied in Harris’s antelope squirrels. Animals were not chemically immobilized due to the nature of the procedures (i.e., momentary pain). Lack of chemical immobilization also limits time spent handling, risk of thermoregulatory distress, and additional stress caused by immobilization procedures (Sikes & Animal Care and Use Committee, 2016). The DNA of the collected tissue samples was extracted at the University of Arizona Conservation Genetic Laboratory using a Qiagen DNeasy blood and tissue extraction kit (Qiagen Inc., Hilden, Germany). We used a Qubit fluorometer (Invitrogen™, Thermo Fisher Scientific Inc., Waltham, MA, USA) to quantify DNA products before sequencing.

Figure 1 Sampling locations of Harris’s antelope squirrels (A. harrisii) in the Santa Rita Experimental Range, AZ.

Green points indicate sampling sites. Inset map shows the location of the Santa Rita Experimental Range (yellow point) in the state of Arizona (shaded blue). Map data ©2024 Google.

Library preparation and genomic sequencing

DNA (1,100 ng) from each sample was sent to Floragenex (Beaverton, OR, USA) for library preparation and double-digest restriction site associated DNA sequencing (ddRADseq). Library preparation was performed using the restriction enzymes PstI and Msel, with a size selection range of 250–800 base pairs (bp). The final pooled library was sequenced on an Illumina HiSeq 3000 with 1 × 100 bp reads.

Bioinformatic pipeline

We used FASTQC v. 0.11.9 (Andrews, 2010) to confirm presence of the enzyme cut sites and assess read quality. We used the program process_radtags in Stacks v. 2.60 (Catchen et al., 2013) to demultiplex the pool into individuals based on in-line adapter barcodes, quality filter reads (with a minimum Phred score of 30) and remove reads with missing RAD enzyme cut sites. Following the error clean-up, we used Stacks to assemble loci and call single nucleotide polymorphisms (SNPs).

No reference genomes exist for A. harrisii. Thus, we used the Stacks denovo_map pipeline with the output from process_radtags to de novo assemble loci with default parameters. We also applied the—write_random_snp flag to obtain one SNP per locus from the populations module. We sequentially filtered SNPs using PLINK v. 1.90 (Purcell et al., 2007), removing loci genotyped in less than 75% of individuals (—geno 0.25) and loci with a minor allele frequency (MAF) less than 5% (—maf 0.05). We further filtered SNPs to remove individuals with more than 50% missing data at the retained loci (—mind 0.5). No individuals were removed during this additional filtering step; however, we found that our sample set included duplicates of four individuals, which were subsequently removed.

Genetic summary statistics

With our unduplicated dataset containing one SNP per locus, we reran the populations module to generate population-level genetic summary statistics (observed and expected heterozygosity; Ho and HE, respectively), nucleotide diversity (π, considering variant and invariant sites), and inbreeding coefficient (FIS). We performed an additional run of populations using a population map file denoting males and females to compare results between sexes in downstream analyses. To estimate effective population size, we used the software NeEstimator v. 2.1 (Do et al., 2014) with the linkage disequilibrium method and a minor allele frequency cutoff of 0.05.

Genetic structure analyses

To better understand population structure, we visualized our data with the R package pophelper (Francis, 2017) and ran a principal component analysis (PCA) as well as a discriminant analysis of principal components (DAPC) using adegenet (Jombart, 2008). We derived DAPC results based on the K value with the lowest Bayesian Information Criterion (BIC). Additionally, we used ADMIXTURE (Alexander, Novembre & Lange, 2009) to analyze population substructure and to determine the most likely number of ancestral lineages, identifying the best-supported K value by the lowest cross validation error.

Relatedness

To determine whether dispersal is female biased, we tested whether average female-female relatedness differed from male-male relatedness or female-male relatedness. We used the related package in R (Pew et al., 2015) to estimate dyadic relatedness (the dyadml estimator) between individuals. To ensure our data met test assumptions, we assessed the distribution of relatedness. Finding it non-normal, we used a Kruskal-Wallis test to determine whether male-male, male-female, and female-female relatedness are derived from the same distribution (i.e., whether any group is statistically different from another). We then used a Dunn post-hoc test with a Bonferroni correction using FSA (Ogle et al., 2023) to identify which specific group was different from the others.

Isolation by distance

We tested whether genetic distance and geographic distance were significantly correlated (isolation by distance; IBD) to determine whether higher calling propensity observed in female squirrels is due to higher relatedness between neighboring females. We performed a Mantel test (Mantel, 1967) with the adegenet package (Jombart, 2008), using geographic Euclidean distance and the inverse proportion of shared alleles (DPS) for each sample pair and 9,999 permutations to assess significance. The Mantel test is used to evaluate whether genetic and geographic distance are linearly correlated and is a widespread technique used to assess spatial structure in genetic data (Diniz-Filho et al., 2013; Quilodrán, Currat & Montoya-Burgos, 2023). We further ran the test separately for males (n = 18) and females (n = 29) to determine if this relationship differed between sexes. To explore any potential non-linear patterns between genetic and geographic distance, we ran a Mantel correlogram analysis using the vegan package (Diniz-Filho et al., 2013; Oksanen et al., 2022). We ran separate correlogram analyses for male-male and female-female genetic distance, binning geographic data into six distance classes with break points at 0, 100, 500, 1,000, 3,000, and 6,000 m.

Results

Genetic diversity estimates

We retained 21,958 SNPs following quality filtering and the removal of four duplicate individuals (n = 47 individuals, 1,081 dyads). Genetic summary statistics indicated an inbreeding coefficient of 0.079 suggesting that a low level of inbreeding might be occurring within this population. Genetic diversity parameters showed an observed heterozygosity (±SE) of 0.269 ± 0.001 and expected heterozygosity of 0.286 ± 0.001. Furthermore, sampled individuals showed nucleotide diversity (pi) of 0.003 (considering variant and invariant sites). Effective population size based on NeEstimator calculations using a minor allele frequency cutoff of 0.05 was 105.7 (95% confidence interval 105.5, 105.9).

Population structure

Our population structure results showed that all squirrels sampled fell under one panmictic population. Furthermore, PCA and DAPC results grouped samples under one cluster based on the lowest BIC value (Figs. S1 and S2), indicating that all individuals sampled belong to one population. Admixture results identified K = 1 as the best supported number of clusters, having the lowest cross-validation error (0.55).

Relatedness

Average relatedness (±SE) between individuals was 0.014 ± 0.001 based on the dyadic likelihood estimator (n = 47 individuals). Female-female relatedness (r = 0.021 ± 0.003), female-male relatedness (r = 0.008 ± 0.002) and male-male relatedness (r = 0.015 ± 0.001) were not derived from the same distribution (Kruskal-Wallis test; χ2 = 181.21, df = 2, p < 2.2e−16). Results of Dunn post-hoc test showed that male-male relatedness differed from female-male (Dunn post-hoc; Z = −11.8, p = 1.11e−31) and female-female relatedness (Dunn post-hoc; Z = −13.13, p = 6.6e−39). Female-female relatedness and female-male relatedness also differed (Dunn post-hoc; Z = −2.42, p = 0.046).

Isolation by distance

Genetic distance between squirrels showed a significant relationship with geographic location (Mantel test; R = 0.18, p = 0.001, n = 47), such that squirrels at closer distances were more related (Fig. S3). When squirrels were separated by sex, correlation between genetic and geographic distance strengthened and significant patterns remained for females (Mantel test; R = 0.29, p = 1e−4, n = 29; Fig. 2), but not males (Mantel test; R = 0.03, p = 0.39, n = 18). When geographic distances were grouped into distance classes, female-female genetic distance was positively correlated (Mantel correlogram analysis; R = 0.46, p = 0.001, n = 86) with geographic distance at close distances (0–100 m), but negatively correlated (Mantel correlogram analysis; R = −0.26, p < 0.01, n = 238) at further distances (3–6 km; Table S1). Male-male genetic distance exhibited very low or no correlation with geographic distance (Mantel correlogram analysis; R < 0.1, p > 0.05) regardless of distance class (Fig. S4, Table S2).

Figure 2 Isolation by distance of female and male Harris’s antelope squirrels.

Scatterplot showing the relationship between geographic distance (spatial Euclidean in meters) and genetic distance (inverse proportion of alleles shared between individuals) of (A) female Harris’s antelope squirrels (n = 29) and (B) male Harris’s antelope squirrels (n = 18). Colors represent the relative density of points: red showing higher density, yellow medium density, and blue lower density. A Mantel test showed a positive correlation between geographic and genetic distance for females (R = 0.29, p < 0.001) but not males (R = 0.03, p = 0.39).

Discussion

Harris’s antelope squirrels in the SRER showed genetic diversity reflective of small, isolated and fragmented populations, similar to that of northern and southern Idaho ground squirrels (Garner, Rachlow & Waits, 2005; Barbosa et al., 2021). Our population may exhibit similar genetic diversity due to major physical barriers in all cardinal directions, with cities and major highways positioned to the north and west of the SRER and the Santa Rita Mountains extending from the southern boundary of the SRER to the northeastern boundary.

The sex-dependent relationship between geographic distance and genetic distance, as well as the higher relatedness observed between female-female pairs compared to female-male or male-male pairs provides evidence that A. harrisii exhibits male-biased dispersal, or female philopatry. The results of the Mantel correlogram analysis provide further supportive evidence of male-biased dispersal and spatial clustering of female relatives in our population of A. harrisii, with female squirrels that are further distances apart being largely unrelated. Female philopatry is a widespread mammalian trait, particularly in species with polygynous mating systems (Lawson Handley & Perrin, 2007; Mabry et al., 2013) like those found in ground squirrels. Male-biased dispersal is exhibited in a number of other ground squirrel species (Devillard et al., 2004), including Golden-mantled ground squirrels (Callospermophilus lateralis; Nguyen & Van Vuren, 2024), Columbian ground squirrels (Waterman, 1992; Neuhaus, 2006), rock squirrels (Otospermophilus variegatus; Shriner & Stacey, 1991), and Belding’s ground squirrels (Urocitellus beldingi; Holekamp, 1984). Male-biased dispersal in ground squirrels may help prevent inbreeding (Holekamp, 1984) or reduce exposure to female aggression (Neuhaus, 2006). Relationships between geographic and genetic distance vary across ground squirrel species, however. Speckled ground squirrels (Spermophilus suslicus) show significant positive correlations between genetic and geographic distances (Matrosova et al., 2016), whereas northern and southern Idaho ground squirrels (Urocitellus brunneus and U. endemicus, respectively; Garner, Rachlow & Waits, 2005), round-tailed ground squirrels (Xerospermophilus tereticaudus; Munroe & Koprowski, 2014), and California ground squirrels (Otospermophilus beecheyi; Glover, 2018) do not exhibit positive relationships between genetic and geographic distances. Variation in the relationship between geographic distance and relatedness among squirrel species could be due to differences in how sex-biased dispersal is expressed, which can vary substantially among species (Devillard et al., 2004; Lawson Handley & Perrin, 2007). Although female philopatry is widespread, the magnitude of bias (e.g., whether one sex exhibits complete philopatry or not) and the ratio of dispersal distances between males and females varies among species (see Lawson Handley & Perrin, 2007 for review). Variation in the degree of sex-bias may explain the relatively weak correlation (R = 0.29) between female relatedness and geographic distance. Adult females were captured within close proximity (<300 m) of other adult females of both high (e.g., r = 0.25) and low (e.g., r = 0) relatedness, indicating that clustering among related individuals is relatively relaxed and some female offspring may also disperse depending on resource availability, mate competition, and aggression from conspecifics (Lawson Handley & Perrin, 2007).

Our results support kin selection theory, in which solitary females that exhibit overlapping territories are expected to be related, such that tolerance of neighbors benefits females via indirect fitness (Hamilton, 1964). Our study area exhibited a uniform distribution of resources at a low spatial density, and antelope squirrels in our study area maintain large, overlapping home ranges ranging from ~0.5 to 7 ha (Burnett & Koprowski, 2024). Other solitary species similarly benefit from female philopatry in a number of ways, including allo-parental care (Gilchrist, 2007; Schradin, Vuarin & Rimbach, 2018), territory acquisition (Lutermann et al., 2006; Goodrich et al., 2010; le Roex et al., 2022; Payne et al., 2024) and thermoregulation (Williams et al., 2013). Females in a number of solitary species, including bobcats (Lynx rufus; Janečka et al., 2007; Payne et al., 2024), Amur tigers (Panthera tigris altaica; Goodrich et al., 2010), and brown bears (Ursus arctos; Støen et al., 2005), share home ranges with their daughters, resulting in kin-related spatial structure that could have important repercussions for indirect fitness and social relationships (Støen et al., 2005; de Oliveira et al., 2022). Although female philopatry and kinship theory is useful for understanding the social relationships in some solitary mammals, other solitary species exhibit adaptive social strategies that are maintained by familiarity with neighbors (Siracusa et al., 2019) or reciprocity (Elbroch et al., 2017). For example, North American red squirrels (Tamiasciurus hudsonicus) are highly territorial but exhibit behavioral plasticity, reducing effort spent defending their territory (i.e., emitting territorial vocalizations) and increasing time spent in the nest as familiarity with their neighbors increases over time (Siracusa et al., 2019). Pumas (Puma concolor) cofeed at kill sites with unrelated individuals and maintain social networks via reciprocity (Elbroch et al., 2017). Some species, like North American red squirrels, can exhibit social strategies governed by both kinship and familiarity simultaneously (Walmsley et al., 2023). Thus, a number of ecological factors can select for social structure to evolve across the spectrum of sociality (Makuya & Schradin, 2024).

Our findings indicate that kin selection resulting from high genetic relatedness between neighboring females may be partially responsible for sex differences in alarm calling behavior in A. harrisii (Burnett & Koprowski, 2020), highlighting the nuanced role that ecological patterns like female philopatry can play in species behavior. Kin selection may have a strong evolutionary influence on antipredatory behavior in many ground squirrels, especially those for which alarm vocalizations likely serve as a warning to surrounding relatives and provide predator details (Ackers & Slobodchikoff, 1999). However, A. harrisii is largely solitary and emits alarm vocalizations at a wide range of amplitudes, including very low amplitudes that do not travel across the landscape and would be difficult for neighboring squirrels to detect (personal observation). Additionally, sex bias in calling propensity is nonsignificant under high-risk contexts (e.g., when confined; Burnett & Koprowski, 2020), indicating that while females may have a lower threshold for calling than males, both sexes vocalize in response to immediate threats. Alarm vocalizations in this species are thought to be directed toward the predator (Burnett & Koprowski, 2020), thus predation pressure may still be primarily responsible for the maintenance of alarm vocalizations in this species. Although these results show alarm calling behavior in our population of antelope squirrels is likely subject to kin selection, whether alarm vocalizations serve multiple functions is still unclear. Alarm calls were frequently given by multiple individuals (Burnett & Koprowski, 2020), but whether alarm calls elicit further alarm calling by neighboring squirrels or whether squirrels were simply calling in response to the same threat is unknown. During our field observations over the course of 2 years, we did not observe a clear behavioral response to alarm calls from neighboring antelope squirrels, however, relatives may benefit from alarm vocalizations if predators leave the immediate area to hunt elsewhere (Blumstein et al., 1997; Zuberbühler, Jenny & Bshary, 1999; Isbell & Bidner, 2016). Although kin selection is an important component underlying the evolution of alarm signals, other evolutionary pressures such as those imposed by the environment and the intended receiver (e.g., conspecifics, predators, or both) are also important to take into consideration (Guilford & Dawkins, 1991; Patricelli & Hebets, 2016).

Conclusions

Our study aimed to determine whether female philopatry influences alarm calling behavior in Harris’s antelope squirrels. We hypothesized that relatedness between squirrels would be correlated with geographic distance and that females would be more closely related to neighboring squirrels than males, based on previous findings of greater calling propensity in female antelope squirrels. We found that dispersal in our population of Harris’s antelope squirrel is male-biased, and genetic distance and geographic distance were positively correlated for female squirrels but not males, indicating that kin selection resulting from female philopatry may be responsible for sex differences in calling behavior. Relatedness between neighboring females supports kin selection theory predicting that solitary females with overlapping home ranges are likely to be related. Our results complement other studies showing that female philopatry plays an important role in the formation of sociospatial patterns in solitary mammals. We additionally found low genetic diversity in our population, suggesting that our population may be somewhat isolated from other populations due to topographical barriers. Our findings add to the body of literature underlining how sociospatial organization may shape the behavior of solitary mammals and highlight the subtle ways that kinship and genetic structure influence the social landscape experienced by solitary mammals.

Supplemental Information

Supplemental Information 1 Principal Component Analysis (PCA) showing genetic overlap between sampling regions.

Principal Component Analysis showing that all four trapping locations likely sampled animals within a single population.

Supplemental Information 2 Value of Bayesian Information Criterion (BIC) with increasing number of clusters.

The number of clusters indicated by the lowest BIC suggests that all individuals can be grouped genetically into a single cluster.

Supplemental Information 3 Isolation by distance of all sampled Harris’s antelope squirrels.

Scatterplot showing the relationship between geographic distance (spatial Euclidean in meters) and genetic distance (inverse proportion of alleles shared between individuals) of all sampled Harris’s antelope squirrels (n = 47). Colors represent the relative density of points: red showing higher density, yellow medium density, and blue lower density. A Mantel test showed a significant relationship between geographic and genetic distance (R = 0.18, p < 0.001).

Supplemental Information 4 Mantel correlogram showing the correlation between geographic distance and genetic distance over distance class for A) female and B) male Harris’s antelope squirrels.

Mantel correlogram showing how the correlation between geographic and genetic distance change with distance class. Female antelope squirrels exhibited a positive correlation between geographic and genetic distance at close distances and a negative correlation at larger distances. Males showed very low correlations between geographic and genetic distance across distance classes. Black points represent significant (p < 0.05) Mantel correlations; white points represent nonsignificant Mantel correlations. Note the difference in scale on the y-axis.

Supplemental Information 5 Results of Mantel correlogram analysis for female Harris’s antelope squirrels.

Summary of Mantel correlogram analysis showing how the relationship between female-female genetic distance and geographic distance changes with distance class. Bold values indicate significance. Class index represents the median of each distance class.

Supplemental Information 6 Results of Mantel correlogram analysis for male Harris’s antelope squirrels.

Summary of Mantel correlogram analysis showing how the relationship between male-male genetic distance and geographic distance changes with distance class. Class index represents the median of each distance class.

Supplemental Information 7 ARRIVE 2.0 Checklist.

We thank our field assistants for their help in the field, including A. Blair, M. Bethel, N. Bokanoski, H. Yomantas, and A. Blanche. We also thank K. Bennett, M. Merrick, and V. Greer for their assistance with training and logistics. Any use of trade, firm, or product names is for descriptive purposes only and does not imply endorsement by the U.S. Government.

Additional Information and Declarations

Competing Interests

The authors declare that they have no competing interests.

Author Contributions

Alexandra Burnett conceived and designed the experiments, performed the experiments, analyzed the data, prepared figures and/or tables, authored or reviewed drafts of the article, and approved the final draft.

Michelle Hein analyzed the data, prepared figures and/or tables, authored or reviewed drafts of the article, and approved the final draft.

Natalie Payne conceived and designed the experiments, analyzed the data, prepared figures and/or tables, authored or reviewed drafts of the article, provided training for laboratory protocols during study, and approved the final draft.

Karla L. Vargas conceived and designed the experiments, analyzed the data, prepared figures and/or tables, authored or reviewed drafts of the article, provided training for laboratory protocols during study, and approved the final draft.

Melanie Culver conceived and designed the experiments, authored or reviewed drafts of the article, provided training for laboratory protocols during study, and approved the final draft.

John L. Koprowski conceived and designed the experiments, authored or reviewed drafts of the article, and approved the final draft.

Animal Ethics

The following information was supplied relating to ethical approvals (i.e., approving body and any reference numbers):

The University of Arizona Institutional Care and Use Committee approved all procedures used during this study.

Vertebrates were captured by University of Arizona personnel under their live capture permits UA IACUC 16–169 and AZGFD SP501610 & SP611944.

DNA Deposition

The following information was supplied regarding the deposition of DNA sequences:

The sequences are available are NCBI SRA: PRJNA1153059.

Data Availability

The following information was supplied regarding data availability:

The data (including raw data and source files for code) and code for analysis is available on Open Science Framework: Burnett, Alexandra, and Karla L. Vargas. 2025. “AMHA Genetics.” OSF. January 10. doi: 10.17605/OSF.IO/WV86C.

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
