# Peer review of "Female philopatry may influence antipredatory behavior in a solitary mammal"

_PeerJ, doi:10.7717/peerj.18933_

## Round 0.1 · original submission · Major Revisions

Dear Authors,

I am pleased to observe that the revisions made have significantly improved the manuscript. However, the reviewers have suggested some major revisions that still need to be addressed. I look forward to receiving your revised version soon, so the manuscript can move forward toward acceptance.

Best regards,
Armando Sunny

Reviewer 1 ·

Basic reporting

The article was generally clearly written and the code appears to be well-documented. I had no issues with the structure or presentation of the article, though it may be worth trying to enhance the resolution of the figures. Overall, I think that the findings are a valuable addition to the field, conditional on more careful interpretation relative to alarm-calling behavior.

Context – Additional discussion of key ideas was sometimes necessary. For example, what factors might explain differences in local relatedness across species (233-239)? How large are antelope squirrel home ranges (257)? How can a lack of sex-bias in alarm calling under high-risk contexts be explained by predation pressure (286-289)?

Sampling locations – More information is needed to understand the sampling locations. Apologies if I have simply missed this information, but it was not apparent to me. While the map helps, it was not clear why these specific areas were selected. This is crucial as it relates directly to the main findings of study, and appears to be the principal driver of measured variation in geographic distances, which are strongly clustered (Figure 2). Are these sites distinct areas of high density, or simply areas that were sampled in an otherwise fairly homogenous distribution of animals? This clustering is not necessarily a problem, but more context on the spatial scale of the results and possible effects of relatedness need to be provided.

Experimental design

I had no qualms with the reproducibility or ethics of the methods, though some procedures could be explained briefly in the results (emphasizing how they will aid in interpretation of biological processes) so that the reader understands why they are being used.

Though not absolutely necessary, I would consider applying models that better capture links between distance and genetic relatedness (e.g., relationship for female relatedness and geographic distance looks fairly non-linear across ~6km). Alternatively one could consider using log-transformed distance, which might better capture what differences in distance mean biologically (e.g., 10 vs. 100m is a more important difference than 3010 vs. 3100m).

Can you explain the purpose of the Dunn post-hoc test, and why it was used in conjunction with the KW test? It would be useful to include this both in the methods and results.

Validity of the findings

The authors present useful findings on the dispersal patterns of Harris’s antelope squirrel. I have no major qualms with the validity of this finding, and think that this result will be of value to the literature, particularly for understanding variation in sex-biased dispersal and local relatedness across mammals.

Mismatch between study and conclusions – The most important issue in this manuscript is the gap between the theoretical focus (alarm calling and kin selection) and the analytic focus (spatial patterns of relatedness). For example, the current first paragraph of the introduction makes me think that the study will be on alarm calling. While it is appropriate to raise connections between these ideas in the Discussion (278-280), the manuscript should be significantly reshaped to emphasize theory on dispersal and local relatedness. Answering the stated question would require additional analyses and data on alarm calling and its effects on conspecifics. Or perhaps, one could test whether females with increased local relatedness tend to produce more alarm calls.

Variation in local relatedness – It would be very interesting to present results on variation in local relatedness across individuals. For example, do most females end up in similar social environments from a relatedness perspective, or are some females surrounded by kin while others are much more isolated? I think that this would be a nice addition that could be discussed in the context of kin selection in the results, i.e., allowing for a reduced emphasis on alarm calling. This could be presented in a figure, or simply reported, depending on the findings. See Croft et al., 2021 for similar ideas about variation in relatedness over time. https://royalsocietypublishing.org/doi/full/10.1098/rspb.2021.1129

Additional comments

182 – Could also report number of dyads here.

77 – Framing a bit too strong here, I’m not sure that these data can provide a test of kin selection.

264 – Extra period

267-268 Nice discussion – important to note though that kinship- and familiarity-based social strategies can operate simultaneously, see Walmsley et al., 2023 https://royalsocietypublishing.org/doi/full/10.1098/rspb.2022.1569

273-275 – See recent review on consequences of social interactions in solitary mammals, Makuya and Schradin 2024 https://zslpublications.onlinelibrary.wiley.com/doi/full/10.1111/jzo.13145

313-315 – Some additional information would be interesting here. Do alarm calls elicit other alarm calls? Do nearby individuals tend to respond behaviorally?

Figure 1 – Figure looks a little blurry, if possible try to save a version in higher resolution for the final MS.

Figure 2 – There seems to show a relatively poor fit between the simple relationship implied by the Mantel test and observed data. I’m not sure exactly how the genetic distance measures would translate to relatedness values, but it looks like closer relatives are generally found within 1km of one another (and several hundred m), but that relatedness and geographic distance are relatively unrelated beyond ~3km.

·

Basic reporting

The English language used throughout the manuscript is professional and clear. A few sentences sound a bit wordy, I have indicated those in the attached PDF.

The literature references used seem appropriate and sufficient.

The article structure is also appropriate for the submitted manuscript.

The raw data (list of occurrence points) and other files were NOT shared or are not available to the reviewer.

The manuscript is self-contained with relevant results to the hypotheses set up. However, the relevance of the hypotheses is somewhat questionable. It appears quite obvious that individuals of the same species and same population are more closely related to each other then individuals found further away. This does not seem like a ground-breaking and novel research question to pursue.

Some additional comments were included in the attached PDF.

Experimental design

The article seems to be original primary research within the aims and scope of the journal.

The manuscript is self-contained with relevant results to the hypotheses set up. However, the relevance of the hypotheses is somewhat questionable. It appears quite obvious that individuals of the same species and same population are more closely related to each other then individuals found further away. This does not seem like a ground-breaking and novel research question to pursue.

As described in more detail in the attached PDF, the sampling numbers and distribution appear to be relatively small and not representative of the species. The authors included 46 individuals who were all distributed within a radius of approximately 3-4km and appeared to be of the same population. For more robust results, I suggest including a higher number of samples, from a larger geographic distribution, and other populations.

No response was provided for the IACUC approval number or ASPA license type and number.

I would also include the sample size in the methods rather than only in the results.

Validity of the findings

The underlying data was not provided to me.

Replication is hardly achievable given that the utilized data, code, and samples are not available.

The achieved results provide a trend but they are arguably not very robust (see experimental design section) and the assessment metrics (e.g. R of 0.29 and 0.03 for the relationship between geographic and genetic distance) are not very indicative.

Some supplemental figures were also described quite poorly.

Impact and novelty were sound and well-defined.

Conclusions were well stated and linked to the research question and supporting results.

Additional comments

Thank you very much for submitting your manuscript to PeerJ.

I find your research highly interesting, yet I believe that it needs some more work and sampling effort to achieve robust results.

·

Basic reporting

The authors have done an excellent job of preparing this manuscript. From their introduction to their discussion, they have presented a clear direction for their study with adequate context and background to allow interpretation of their results. Their methods are sound, clearly described, and appropriately interpreted. The only comments I have on how to improve this manuscript would be:

1) To include the trends seen in males in Figure 2 as well. While I don't think this change is absolutely necessary, a significant portion of your discussion centres around the sex differences in this species, so highlighting these differences in a figure would be helpful to readers.
2) Mentioning where your raw data can be accessed within the manuscript.

Experimental design

no comment

Validity of the findings

no comment

Additional comments

I thank the authors for their submission and the thoroughly enjoyable manuscript. I think they have done a fantastic job and that their findings are of significant value to the current literature and warrant publication. Other than my minor comments listed above, I think this article is ready for publication.

---

## Round 0.2 · Major Revisions

Dear Author's,

Thank you for submitting your manuscript, "Female Philopatry as Evidence of Kin Selection in a Solitary Mammal,". One of the reviewers has raised concerns regarding the statistical analyses used in your study. We kindly request that you address these issues by providing further details and justifications for your statistical methods and ensuring that all assumptions are met.

Best regards,

Armando Sunny

Reviewer 1 ·

Basic reporting

No comment

Experimental design

No comment

Validity of the findings

No comment

Additional comments

Generally, I was very impressed with how thoroughly the authors addressed the comments and think that the article should proceed to publication.

There is one important remaining issue, which is that the evidence for kin selection is overstated in the Title and Abstract. As the authors note in the Discussion, these findings only indicate that kin selection “may be partially responsible for sex differences in alarm calling behavior”.

In my view, (and seemingly the authors’ at points), there is no (strong) evidence for kin selection in this system. The cited 2020 paper makes clear that males also use alarm calls, and even as often as females in some behavioral contexts, suggesting that other mechanisms are involved. More importantly, the authors of the present article note that there has been no evidence of behavioral responses to the alarm calls of neighbors, raising the question of whether these calls actually benefit conspecifics in the first place.

The authors mention that they have had difficulty collecting adequate genetic samples linked to behavioral observations. This is very understandable, and unproblematic for the present analysis. Still, the title and abstract should more closely reflect the findings, as is done very nicely in the Discussion.

·

Basic reporting

Thank you for thoroughly responding to my comments and mostly implementing my suggestions. The revised manuscript appears to have noticeably improved through this first revision. There are still a couple of points I would like to outline that raised my concern when reviewing this manuscript:

1) I did not find your statements on how you sufficiently assuredly fulfilled all statistical assumptions of the chosen statistical test (Mantel test and others) anywhere. I highly encourage you to include those statements. In most ecological studies, statistical assumptions are far from met, making the results untrue and useless. I hope you sufficiently fulfill them.

2) Still on a similar topic to my first comment, I highly recommend reading the following manuscript (https://doi.org/10.1111/2041-210x.12018) and rebuttal in your manuscript how your study does not fall within this category of producing erroneously low p-values.

3) Continuing the thought from above, I encourage you to move away from p-values-based statistical approaches, especially in ecology. The statistical assumptions are sheer impossible to meet, and without those, the results cannot really be utilized or evaluated as meaningful. From an academic standpoint, in ecology, utilizing ‘inference from prediction‘ approaches usually performs more robustly and gives you more freedom in your data and approach. I recommend reading the following resources
- Breiman, L. Random forests. Machine learning 45, 5–32 (2001).
- Breiman, L. Statistical modeling: The two cultures (with comments and a rejoinder by the author). Stat. Sci. 16, 199–231 (2001).
- Huettmann, F., Andrews, P., Steiner, M., Das, A. K., Philip, J., Mi, C., ... & Barker, B. (2024). A super SDM (species distribution model) ‘in the cloud’ for better habitat-association inference with a ‘big data’ application of the Great Gray Owl for Alaska. Scientific Reports, 14(1), 7213.

4) If possible, I also recommend publishing the occurrence points of your sampling locations on GBIF.org, to contribute to the citizen-science effort.

5) I understand your reasoning for the sample location and small sample size (scope of funding, etc.). For this reason, in my view, it is essential to use appropriate language when stating your results. Because with this sample size, you can only 'contribute to a growing body of evidence' instead of ultimately proving the hypothesis. It seems like this is clear to you; I encourage you to go through your manuscript and adjust the wording as needed to ensure the scope and meaning of the findings are transparent to the readers.

6) I also still lack ISO-compliant metadata, which is supposed to accompany your data. I highly encourage you to create it and include it in your final submission.

Experimental design

The revised manuscript fulfills all experimental design requirements (apart from those mentioned in the basic reporting section above).

Validity of the findings

The revised manuscript fulfills all validity of findings requirements (apart from those mentioned in the basic reporting section above).

Additional comments

There are no additional comments.

---

## Round 0.3 · accepted · Accept

Dear Authors:

We are pleased to inform you that your manuscript titled "Female philopatry may influence antipredatory behavior in a solitary mammal" has been accepted for publication in PeerJ.

Thank you for choosing PeerJ as the platform to share your work. We look forward to seeing the positive impact of your research on this important topic.

Best regards,
Armando Sunny

Reviewer 1 ·

Basic reporting

no comment

Experimental design

no comment

Validity of the findings

no comment

Additional comments

The authors have addressed my main comment from the previous iteration, in that the Title now better matches how the results of the study are interpreted. Given that the title is somewhat general (referring to a "solitary mammal"), I would make sure to include the species name as a keyword if possible.

I was also glad to see that the authors did not take on the suggestion to move towards “inference from prediction” approaches, which in my view are unsuitable for the goals of the study.

Otherwise I have no further comments and think that the paper will be a valuable addition to the field. I also take the point that alarm calling may be beneficial purely via behavioral impacts on predators, which I had not considered.

·

Basic reporting

The authors have sufficiently responded to my comments and implemented my suggestions. Therefore, I recommend this manuscript for publication.

Experimental design

The authors have sufficiently responded to my comments and implemented my suggestions. Therefore, I recommend this manuscript for publication.

Validity of the findings

The authors have sufficiently responded to my comments and implemented my suggestions. Therefore, I recommend this manuscript for publication.